# Urine Proteome in Distinguishing Hepatic Steatosis in Patients with Metabolic-Associated Fatty Liver Disease

**DOI:** 10.3390/diagnostics12061412

**Published:** 2022-06-07

**Authors:** Chang-Hai Liu, Shanshan Zheng, Shisheng Wang, Dongbo Wu, Wei Jiang, Qingmin Zeng, Yi Wei, Yong Zhang, Hong Tang

**Affiliations:** 1Center of Infectious Diseases, West China Hospital, Sichuan University, Chengdu 610041, China; changhai.liu@hotmail.com (C.-H.L.); dongbohuaxi@163.com (D.W.); 17761215287@163.com (W.J.); qmzeng94@163.com (Q.Z.); 2Division of Infectious Diseases, State Key Laboratory of Biotherapy and Center of Infectious Disease, West China Hospital, Sichuan University, Chengdu 610041, China; 3Key Laboratory of Transplant Engineering and Immunology, MOH, West China-Washington Mitochondria and Metabolism Research Center, West China Hospital, Sichuan University, Chengdu 610041, China; janezss@163.com (S.Z.); shishengwang@wchscu.cn (S.W.); 4Department of Radiology, West China Hospital, Sichuan University, Chengdu 610041, China; drweiyi057@163.com; 5Institutes for Systems Genetics, West China Hospital, Sichuan University, Chengdu 610041, China

**Keywords:** metabolic-associated fatty liver disease, proteomics, hepatic steatosis

## Abstract

Background: In patients with metabolic-associated fatty liver disease (MAFLD), hepatic steatosis is the first step of diagnosis, and it is a risk predictor that independently predicts insulin resistance, cardiovascular risk, and mortality. Urine biomarkers have the advantage of being less complex, with a lower dynamic range and fewer technical challenges, in comparison to blood biomarkers. Methods: Hepatic steatosis was measured by magnetic resonance imaging (MRI), which measured the proton density fat fraction (MRI-PDFF). Mild hepatic steatosis was defined as MRI-PDFF 5–10% and severe hepatic steatosis was defined as MRI-PDFF > 10%. Results: MAFLD patients with any kidney diseases were excluded. There were 53 proteins identified by mass spectrometry with significantly different expressions among the healthy control, mild steatosis, and severe steatosis patients. Gene Ontology (GO) and the Kyoto Encyclopedia of Genes and Genomes (KEGG) analyses of these significantly changed urinary molecular features correlated with the liver, resulting in the dysregulation of carbohydrate derivative/catabolic/glycosaminoglycan/metabolic processes, insulin-like growth factor receptor levels, inflammatory responses, the PI3K–Akt signaling pathway, and cholesterol metabolism. Urine alpha-1-acid glycoprotein 1 (ORM1) and ceruloplasmin showed the most significant correlation with the clinical parameters of MAFLD status, including liver fat content, fibrosis, ALT, triglycerides, glucose, HOMA-IR, and C-reactive protein. According to ELISA and western blot (30 urine samples, normalized to urine creatinine), ceruloplasmin (ROC 0.78, *p* = 0.034) and ORM1 (ROC 0.87, *p* = 0.005) showed moderate diagnostic accuracy in distinguishing mild steatosis from healthy controls. Ceruloplasmin (ROC 0.79, *p* = 0.028) and ORM1 (ROC 0.81, *p* = 0.019) also showed moderate diagnostic accuracy in distinguishing severe steatosis from mild steatosis. Conclusions: Ceruloplasmin and ORM1 are potential biomarkers in distinguishing mild and severe steatosis in MAFLD patients.

## 1. Introduction

Non-alcoholic fatty liver disease (NAFLD), characterized by ectopic fat deposition in the liver, affects up to 30% of the worldwide population [1,2,3]. The overall prevalence of NAFLD in China over the past two decades reached 29.6%, accounting for more than 20% of the global NAFLD population [4,5]. The spectrum of NAFLD extends from liver steatosis to non-alcoholic steatohepatitis (NASH), and sequentially extends to liver fibrosis, cirrhosis, or hepatocellular carcinoma. Despite the gains in our understanding of NAFLD/NASH over the past two decades, some dissatisfaction has been proposed with the name of “non-alcoholic”, which overemphasizes “alcohol” and underemphasizes metabolic risk factors [6]. A name changing from NAFLD to metabolic-associated fatty liver disease (MAFLD) has been suggested [7]. Of note, NAFLD was more defined as a biopsy-proven diagnosis disease; however, in the new definition of MAFLD, the majority of patients with hepatic steatosis have been detected firstly by imaging techniques or biomarkers, combined further with being overweight, obesity, and type 2 diabetes mellitus (T2DM), which could directly result in a diagnosis of MAFLD. Therefore, MAFLD is more focused on metabolic factors and obesity, rather than a biopsy-proven diagnosis.

Hepatic steatosis is the first step of MAFLD diagnosis, and it is a risk predictor that independently predicts insulin resistance, cardiovascular risk, and mortality [8,9,10,11,12,13]. Liver biopsy is still the gold standard to assess liver steatosis, ballooning, and lobular inflammation, as in previous NAFLD and NASH diagnoses. However, the limitations of a biopsy, such as the associated risks, sampling errors, invasiveness, and poor acceptability, are driving studies towards non-invasive methods. On the other hand, the diagnosis of MAFLD emphasizes a diagnosis based on three alternative methods, i.e., “detected either by blood biomarkers/scores, imaging techniques or by liver histology”. Therefore, serum- and image-based methods were recently proposed in multiple studies for the screening and diagnosis of mild/severe hepatic steatosis and NAFLD/MAFLD [14,15,16,17,18,19,20,21].

Magnetic resonance imaging (MRI), which measures the proton density fat fraction (MRI-PDFF), is a new quantitative imaging technique that enables the accurate and precise assessment of the liver triglyceride content and steatosis over the entire liver [15,22]. Mild hepatic steatosis is defined as MRI-PDFF 5–10%, and severe hepatic steatosis is defined as MRI-PDFF > 10% [15,22]. Hepatic fat content > 10% is defined as MRI-PDFF > 10% as this threshold has been used in several therapeutic trials as an inclusion criterion [15]. Of note, MRI-PDFF has been proposed and generally accepted as an endpoint in several early-phase clinical trials [15,23,24,25] and a gold standard in observational studies [26,27]. However, challenges including potential radiation exposure risks, the demand for technical expertise, and a high time/labor costs (30–60 min per patient) have restricted the use of MRI-PDFF in screening patients with MAFLD in general clinical practice.

In proteomics studies, urine represents many frequently used biomarkers in monitoring and diagnosing human diseases, due to its accessibility. Urine samples have the additional advantage of easily reflecting dynamic changes in disease or health conditions, resulting in a better screening and diagnosis biomarker that can predict the progression of diseases [28,29]. Recently, mass spectrometry diagnosis of NAFLD/MAFLD has been utilized to study proteomics data of liver tissue and serum [11,30,31,32,33,34,35]. However, little is known about whether urine samples can reflect the altered metabolic/lipid features in the liver and the severity of hepatic steatosis. There is an urgent need for reliable, low-cost, and low-invasiveness diagnostic techniques using a urine sample to differentiate severe hepatic steatosis patients from mild hepatic steatosis patients.

The aim of the present study was to investigate whether the urine molecular pattern could reflect the pathobiochemistry for MAFLD patients who were classified as having mild hepatic steatosis or severe steatosis by MRI-PDFF. Next, we aimed to explore the urine proteins most strongly correlated with hepatic steatosis by information analysis. Lastly, an independent validation cohort was established to validate the potential urine biomarker that could non-invasively diagnose mild and severe hepatic steatosis in MAFLD patients by using western blot and ELISA.

## 2. Method

### 2.1. Patients and Urine Sample Collection

Patients were diagnosed in the hepatologic disease center of West China Hospital of Sichuan University. The inclusion criteria were as follows: (1) patients 18 to 39 years old; (2) patients diagnosed with MAFLD according to the guideline [7,36], which is described in detail in the following paragraph; and (3) MAFLD patients without any other kind of disease, such as cancer or inflammatory diseases, etc. The exclusion criteria were as follows: (1) patients with missing clinical parameters; (2) patients with renal insufficiency (creatinine level of >1.5 mg/dL in men or >1.4 mg/dL in women) or a previous diagnosis of any kind of chronic or acute kidney disease; (3) patients who had received treatment for MAFLD; and (4) patients with contraindications to MRI, extreme claustrophobia, weight or girth exceeding MRI scanner capability. The discovery set of 27 participants were recruited and their urine samples were collected from May 2020 to July 2020, and the validation set of 30 participants was recruited and their urine samples were collected from August 2020 to October 2020. Patients were divided into 3 groups: healthy control group, mild hepatic steatosis group, and severe hepatic steatosis group. MRI-PDFF was used to measure mild hepatic steatosis defined as MRI-PDFF 5–10% and severe hepatic steatosis was defined as MRI-PDFF > 10%. The study protocol conformed to the ethical guidelines of the 1975 Declaration of Helsinki. The Institutional Review Board Committee of West China Hospital of Sichuan University approved the study protocol. The study was performed by following the ethical guidelines expressed in the Declaration of Helsinki and the International Conference on Harmonization Guidelines for Good Clinical Practice. Informed consent was obtained from all subjects.

The diagnosis was in line with MAFLD criteria [7,36]. In detail, the diagnosis of MAFLD is based on recognizing underlying alterations in metabolism, beyond the histological classification of NAFLD. MAFLD is defined by the presence of steatosis (by histology or imaging) and overweight or at least two metabolic risk factors: overweight/obesity, diabetes mellitus, or metabolic dysfunction. Metabolic syndrome was defined regarding the presence of at least two of the following symptoms: (1) waist circumference of >102 cm for males or >88 cm for females, (2) hypertension defined as arterial blood pressure of ≥130/85 mmHg, as well as those under anti-hypertension therapy, (3) hyperlipidemia (triglyceride (TG) ≥ 1.70 mmol/L or under specific therapy for hyperlipidemia), (4) low high-density lipoprotein cholesterol (HDL-C) level (<1.0 mmol/L for males or <1.3 mmol/L for females), (5) diagnosis of prediabetes, or (6) a hypersensitive C-reactive protein level of >2 mg/L [7].

Patients with other liver diseases, such as viral hepatitis and autoimmune liver disease, were excluded. Participants with excessive alcohol intake were excluded. Of note, patients with any kind of kidney disease (GFR, albumin excretion) were excluded to reflect the natural characteristics of steatosis. The (Controlled Attenuation Parameter) CAP, MRI-PDFF, and laboratory tests were used for healthy controls to exclude MAFLD and any other metabolic-associated diseases, including hypercholesteremia, hypertriglyceridemia, abnormal transaminase, etc. Similarly, healthy controls with liver disease, cancer, excessive alcohol intake, kidney disease, and any other disease were excluded. FIB-4 was calculated as: age ([yr] × AST [U/L])/((PLT [10^9^/L]) × (ALT [U/L])^1/2^) [37].

Urine samples were obtained on the day of performing MRI-PDFF. As urine samples are diluted simply by drinking water, all the urine samples were collected when the participants visited the clinic after dry fasting for >8 h. The midstream of the morning urine was obtained for this study. Urine was centrifuged (1800× *g* for 10 min) to remove debris and was stored frozen at −80 °C. All the urine sample collection and storage conditions followed the same procedure, as follows.

### 2.2. MRI-PDFF-Measured Liver Fat Content

For MRI-PDFF, advanced magnetic resonance imaging (MRI)-based phenotyping was performed at the UCSD MR3T Research Laboratory using the 3T research scanner (GE Signa EXCITE HDxt; GE Healthcare, Waukesha, WI, USA), with all participants in the supine position, in the Radiology Department of West China Hospital. MRI-PDFF was used to measure mild hepatic steatosis, defined as MRI-PDFF 5–10%, and severe hepatic steatosis was defined as MRI-PDFF > 10%. The details of the MRI protocol have been previously described in references [15,26].

### 2.3. Controlled Attenuation Parameter (CAP) and Liver Stiffness Measurements (LSM)

Liver steatosis was defined by the controlled attenuation parameter, and liver fibrosis was assessed via liver stiffness measurements, performed through transient elastography (FibroScan^®^EchoSens, Paris, France), which is most frequently used in patients with fatty liver disease [38,39]. The stratification of the severity of steatosis was as S0–3, and the cut-off of CAP for steatosis was 248 for S1, 268 for S2, and 280 for S3, respectively [40]. Significant liver fibrosis ≥ F1 [39].

### 2.4. Proteomics Sample Preparation and LC–MS/MS Analysis

All 27 urine samples were analyzed by LC–MS/MS in one batch with randomization and were double-blinded between the sample collector and manipulator/analyzer of LC–MS/MS. All the urine sample collections and storage conditions followed the same procedure described in the following. Human urine proteomics samples of 1 mL were collected and centrifuged at 2000× *g* for 4 min to remove cell debris. The supernatant was transferred into a 10 kDa ultrafiltration tube (Merk Millipore, Tullagreen, Carrigtwohill, Co., Cork, Ireland) and washed twice with 200 μL UA buffer (8 M urea, 0.1 M Tris, pH 8.5). The urinary proteins were resuspended with 200 μL UA buffer containing 20 mM dithiothreitol (DTT) and incubated at 37 °C for 16 h. The mixture was then alkylated by 50 mM iodoacetamide (IAA) in the dark at room temperature for 30 min. After this, the samples were washed three times with 200 μL of 50 mM ammonium bicarbonate (NH_4_HCO_3_) by centrifugation at 13,000× *g* for 15 min at room temperature. Then, proteome samples were digested with 1 μg of trypsin (Promega; Madison, WI, USA) and Lys-C (Promega; Madison, WI, USA), and incubated at 37 °C for 14 h. Both trypsin and Lys-C were added simultaneously to reduce the number of missed cleavages. The filter tubes were washed twice with 100 μL of water by centrifugation at 13,000× *g* for 15 min at room temperature. The flow-through fractions were collected. The peptide concentration was determined using a quantitative colorimetric peptide assay kit (Thermo Fisher Scientific, Waltham, MA, USA) based on absorbance at a wavelength of 480 nm. The digested peptides were dried under vacuum before LC–MS/MS analysis. Moreover, the digested peptides were normalized to 500 μg/uL, the same concentration at the peptide level.

Then, 1 μg of each sample was taken for LC–MS/MS analysis on an Orbitrap Exploris 480 coupled with EASY-nLC 1200 (Thermo Fisher Scientific, Waltham, MA, USA) and FAIMS.

The samples were loaded into one column of the capillary device (25 cm multiple by 75 μm), which was packed with C18 reverse phase particles (1.9 μm, Phenomenex, Torrance, CA, USA). The peptides were eluted with a 78 min nonlinear gradient: 3–8% B for 2 min, 8–25% B for 52 min, 25–38% B for 14 min, 38–100% B for 2 min, and 100% B for 8 min. Buffer A contained 0.1% FA in H_2_O, and Buffer B contained 0.1% FA and 80% ACN in H_2_O, with a flow rate of ~300 nL/min.

The MS parameters during detection were as described in the following steps for MS1: an ultra-high-field Orbitrap analyzer was used for the full MS survey scans at a resolution of 60,000 at *m*/*z* 200 over a mass range of *m*/*z* 350–1550. FAIMS CV was set as −45 V, and an automatic gain control (AGC) target value of 3 × 10^5^ with a maximum injection time of 50 ms was used. For MS2, 45 DIA variable windows were acquired at an Orbitrap resolution of 15,000 at *m*/*z* 200. Cycle time for the full scan spectrum and multiple secondary spectra was 3 s, and an AGC target value of 10^5^ with a maximum injection time of 22 ms was used. HCD collision energy was 30%.

DIA–MS data analyses for proteomics were processed using Spectronaut (version 15.0.210615.50606). The parameters were as follows: the raw data files were all directly searched in “directDIA” mode against the Swiss-Prot protein database (September 2020, 20,375 entries). The fixed modification was Carbamidomethyl (C). Variable modifications contained Oxidation (M) and Acetyl (Protein N-term). The other settings were set at the default values, and the filtration of the assigned peptides was performed at the peptide level with 1% false discovery rate (FDR). Identifications were kept with more than two unique peptides, which were 1% FDR and ion score more than 20, restricted to 1% FDR at the protein level. Quantification of all the identified peptides was calculated by peak areas derived from MS2 intensity. The data presented in this study can be obtained via the Proteome Xchange Consortium repository, accession number PXD026333.

### 2.5. Western Blot and ELISA

Urine pretreatment: Briefly, thawed urine was centrifuged (1800× *g* for 10 min) to remove debris before processing. 100μL urine sample was mixed with 1.5 mL of ice-cold acetone methanol (1:1). The mixture was placed at −20 °C for 24 h to obtain protein precipitation. The protein precipitation was dissolved in 100μL lysis buffer (25 mM Tris-HCl, pH 7.6, 150 mM NaCl, 1% NP-40, 1% deoxysodium cholate, and 0.1% SDS) at 30 °C for 1 h.

The concentration of proteins of each sample were quantified by the bicinchoninic acid method (Pierce, Rockford, IL, USA); and 20 ug of total protein were taken for electrophoretic separation on 12% SDS-PAGE gels, and then transferred onto polyvinylidene difluoride membranes and blocked by 5% nonfat dry milk (NFDM). Membranes with protein samples were subsequently reacted with anti-ceruloplasmin (cat. no. ab157452; Abcam, Cambridge, UK), anti-ORM1 (cat. no.16439-1-AP; Proteintech, Wuhan, China) overnight at 4 °C, followed by HRP-conjugated secondary antibody (Jackson ImmunoResearch Laboratories, Inc., Baltimore Pike, PA, USA) for 1 h at room temperature. Both chemiluminescent visualization by an ECL detection system and densitometric analysis by Image Lab Software 5.1 (Bio-Rad Laboratories, Hercules, CA, USA) were carried out to assess the immune signals specific to immunoblots. Both ceruloplasmin and ORM1 were quantified using relative absorbance units and normalized to urine creatinine excretion, as described in previous studies [41,42,43]. Quantification of urinary creatinine concentration was performed using the Jaffe reaction [44].

An ELISA kit was used to measure ceruloplasmin (ELISA Kit, ab110449, Abcam, ILC) and ORM1 (ELISA Kit ab243675, Abcam, ILC) according to the handbook.

### 2.6. Statistics

Reporting of data was according to the most frequently used, with the mean ± standard deviation presented for normal continuous variables and median (interquartile range) for non-normal continuous variables. Otherwise, the frequency was used for discrete variables. The normality of distribution and the homogeneity of variance were checked with shapiro.test and bartlett.test in R, respectively. We used Student’s *t*-test and ANOVA with Bonferroni adjustments for continuous samples, and Fisher’s exact test or chi-square test for the qualitative ones. Non-parametric alternatives (Mann–Whitney U and Kruskal–Wallis tests) were used for non-normal distributions.

Ten patients per group (in both proteomic data and validation cohort) would provide the study with approximately 80% power to detect a 40% difference in each protein level. The power/sample size calculation was based on a 2-sided type I error rate of 0.025 using PASS software (version 11.0.7). We also confirmed the results by using Fisher’s exact test.

The proteins with a missing value ratio above 50% across all samples were removed and imputation was implemented with the k-Nearest Neighbor (KNN) algorithm based on the log_2_ transformed median normalized protein abundances [45]. One-way ANOVA test was performed in R (version 4.0.5, aov function) to identify differential proteins among all groups of samples. The proteins with q-value-corrected *p* values less than 0.05 and a relative fold change of 1.5 folds were considered significant [46]. The principal component analysis (PCA) was performed using the R function prompt. The unsupervised hierarchical clustering was implemented with the heatmap package (https://cran.r-project.org/web/packages/pheatmap/index.html (accessed on 20 May 2021)). Fuzzy c-means clustering of protein profiles was carried out using the Mfuzz package [47]. The selection of cluster number was subjective. When we analyze our data, we manually tried the different cluster number (e.g., 6, 9, and 12); when the cluster number was 12, the best tendency or changed reflected the present dynamic changed of urine proteins. The Gene Ontology (GO) processes and Kyoto Encyclopedia of Genes and Genomes (KEGG) pathway analysis were enriched by the cluster Profiler package [48]. Significant associations were calculated by linear regression with filtered proteins and clinical parameters, including BMI, TG, ALT, AST, glucose, etc. The diagnostic/predictive cut-off of each factor was established by means of the Receiver Operating Characteristic (ROC) curve method at a value that maximized specificity and sensitivity according to the Youden index. The best cut-off value was not determined in the present study due to the limited sample size. R version 4.0.5 and GraphPad Prism version 7 were used for the data analyses.

## 3. Results

### 3.1. Characterization of Urine Proteomes in Healthy Controls and Patients with MAFLD

Five participants were excluded because of missing data. Three participants were excluded because of a reported history of liver disease. Finally, a total sample of 57 subjects were included for analysis, 27 for the discovery proteomics analysis set and 30 for the validation set, respectively. Figure 1A summarizes the sample compilation and statistical analyses. In total, 27 urine specimens passed the quality check (QC), with 7 healthy controls and 20 MAFLD patients without any kind of kidney disease. All patients were subjected to liver fat content measurement with MRI-PDFF; 8 patients were defined as having mild hepatic steatosis, and 12 patients were defined as having severe hepatic steatosis. (Figure 1A,B) Laboratory procedures were all followed to detect sample heterogeneity by the same sample preparation processes, as well as the mass spectrometry analysis. Of the included participants, the BMI, waist circumference, body weight, ALT, TG, total cholesterol, and HDL-cholesterol were significantly different among the healthy control, mild hepatic steatosis, and severe hepatic steatosis groups, while age, male gender, ALP, GGT, glucose, and FIB-4 were not different (Figure 1C).

The number of identified proteins in the control group, mild steatosis group, and severe steatosis group grew quickly and gradually became saturated when the sample size increased (Figure 2A). The protein numbers for the healthy control, mild steatosis, and severe steatosis groups were 2152 ± 187.2, 2552 ± 149.5, and 2404 ± 135.7, respectively. Detailed information about the identified proteins for each sample is presented in Appendix A.

In total, 2087 proteins were commonly identified and were numerically quantified among the healthy controls, mild steatosis patients, and severe steatosis patients (Figure 2A). Among these identified proteins, 304 and 310 proteins were uniquely and specifically expressed in mild steatosis patients and severe steatosis patients, respectively (Figure 2A). The average abundance was nearly the same for the mild and severe steatosis samples compared with the healthy controls (Figure 2B). Principal component analysis (PCA) was further performed to corroborate the previously distinct clusters, resulting in the healthy controls and MAFLD being divided into two groups (Figure 2C).

### 3.2. Urine Proteomics Differentiates Mild/Severe Hepatic Steatosis of MAFLD Patients from Healthy Controls

An obvious distinction of urine proteomes between healthy control, mild hepatic steatosis, and severe hepatic steatosis patients was indicated by the adjusted *p*-value (as q-value) < 0.05 of the ANOVA. There were 53 proteins identified with 331 significantly different expressions among the three groups (Appendix A). We found that mild and severe hepatic patients and healthy control samples could be divided into three categories by using hierarchical cluster analysis (performed only on the 53 proteins significant by ANOVA analysis), suggesting the specific characteristics of molecular conditions between the healthy and MAFLD groups (Figure 3).

GO analysis was performed to find significantly changed urinary molecular features of MAFLD dysregulation, including the carbohydrate derivative catabolic process, the glycosaminoglycan process, the aminoglycan metabolic process, the catabolic process, the inflammatory response, insulin-like growth factor receptors, and GTPase complexes (Appendix A). KEGG analysis also implied that the significantly changed urinary molecular features in MAFLD could relate to the PI3K–Akt signaling pathway and cholesterol metabolism (Appendix A).

To identify specific proteins to distinguish mild from severe hepatic steatosis, the 2087 proteins that were completely shared set by the Venn map (Figure 2A), were clustered into 12 significant distinct clusters with the quantified values through Mfuzz, which was performed randomly according to the tendency of protein expression among the groups [47] (Appendix A). Mfuzz intended to find the cluster of urine protein that was up- or down-regulated from the healthy controls, to mild steatosis and to severe steatosis, of which their expression level could directly reflect the severity of hepatic steatosis and be easier for clinical practice. In our study, cluster three contained consistently up-regulated proteins from healthy controls to mild hepatic steatosis and to severe hepatic steatosis patients, whereas clusters one and eight contained the consistently down-regulated filter panels (Figure 4A). We processed ANOVA and Mfuzz using the full set of proteins parallelly and independently. The ANOVA analysis helped us find 53 significant proteins, but it could not tell us the tendency of proteins, and the Mfuzz analysis could help us obtain the tendencies of protein expressions among the groups (cluster one, three, and eight are the ones we were interested in). Then we combined these results together and identified 15 significantly changed proteins, (Appendix A) including nine unique proteins in the up-regulated panel and six specific proteins in the down-regulated panel (Figure 4B). These filtered proteins were highly associated with liver development, immune system processes, the regulation of immune system processes, carboxypeptidase activity, and GTPase activity (Figure 4B). Moreover, among these 16 proteins, Pearson correlation analysis found that there were 2 of 15 unique proteins, urine alpha-1-acid glycoprotein 1 (ORM1) and ceruloplasmin, that showed the most significant correlation with the clinical parameters of MAFLD status, including liver fat content, fibrosis, ALT, triglycerides, glucose, HOMA-IR, and C-reactive protein (Figure 4C, data expressed as log_2_). Numerically, the alpha-1-acid glycoprotein 1 (ORM1) from cluster three (up-regulated panel) was 4.5-fold (Un-log_2_) higher in mild steatosis, and 7.1-fold higher in severe steatosis. Ceruloplasmin from cluster three (up-regulated panel) was 3.1-fold higher in mild steatosis and 4.8-fold higher in severe steatosis (Figure 4D). The diagnostic accuracy of ORM1 and ceruloplasmin for mild hepatic steatosis (ROC 0.911, 95% CI 0.763–1.000, *p* = 0.008 (sensitivity 87.5% and specificity 85.7%) and ROC 0.964, 95% CI 0.877–1.000, *p* = 0.003 (sensitivity 100.0% and specificity 85.7%)) and severe hepatic steatosis (ROC 0.625, 95% CI 0.366–0.884, *p* = 0.135 (sensitivity 66.7% and specificity 62.5%) and ROC 0.708, 95% CI 0.474–0.943, *p* = 0.123 (sensitivity 58.3% and specificity 87.5%)) was calculated based on the expression intensity of proteomic data (Figure 4E).

### 3.3. Validation by Western Blot and ELISA

Western blot and ELISA (results were normalized to the urine creatinine excretion) were performed to validate the expression of ORM1 and ceruloplasmin in the urine samples of MAFLD patients. In the validation set, 30 cases without any kidney disease or other liver diseases were also included, with 10 healthy controls, 10 mild hepatic steatosis patients, and 10 severe hepatic steatosis patients. Figure 5A shows the baseline characteristics of patients in the validation set, in which the distribution was equal between age (*p* = 0.395), male gender (*p* = 0.451), fasting glucose (*p* = 0.056), etc. In the ELISA test, both ceruloplasmin (healthy control 3.02 ± 1.43 vs. mild steatosis 4.27 ± 1.18 vs. severe steatosis 5.61 ± 1.31, *p* < 0.001) and ORM1 (healthy control 1.19 ± 1.085 vs. mild steatosis 3.41 ± 2.61 vs. severe steatosis 4.68 ± 3.154, *p* = 0.011) showed an increased tendency from healthy controls to mild steatosis and severe steatosis (Figure 5B), and these results were further validated by the western blot (Figure 5C). Moreover, based on the ELISA, ceruloplasmin (ROC 0.78, 95% CI 0.57–0.91, *p* = 0.034, sensitivity 81.5%, specificity 79.5%) and ORM1 (ROC 0.87, 95% CI 0.69–0.94, *p* = 0.005, sensitivity 96.5%, specificity 82.5%) showed moderate diagnostic accuracy in distinguishing mild steatosis from the healthy controls. Ceruloplasmin (ROC 0.79, 95% CI 0.58–0.92, *p* = 0.028, sensitivity 91.5%, specificity 80.5%) and ORM1 (ROC 0.81, 95% CI 0.61–0.91, *p* = 0.019, sensitivity 82.5%, specificity 81.5%) also showed moderate diagnostic accuracy in distinguishing severe steatosis from mild steatosis (Figure 5D).

## 4. Discussion

The present study provides the first urine proteomics data on mild/severe hepatic steatosis according to the measurement of MRI-PDFF in MAFLD patients. Rather than focusing on a single biomarker or a subset of molecular markers, the present proteomics data provide a multiparameter and multistage map of NAFLD and NASH. In the present study, we found that the urine molecular characteristics of hepatic steatosis patients showed dysregulation of the carbohydrate derivative catabolic process, the glycosaminoglycan process, the aminoglycan metabolic process, the catabolic process, the inflammatory response, the insulin-like growth factor receptor level, and GTPase complexes. Furthermore, we demonstrated that urine ceruloplasmin and ORM1 were most associated with liver fat content, fibrosis, ALT, triglycerides, glucose, HOMA-IR, and C-reactive protein. Validation by western blot and ELISA confirmed the expression levels and diagnostic value of urine ceruloplasmin and ORM1.

Urine is more accessible and non-invasive than blood tests; thus, urine has been proposed in multiple research fields as a biomarker of human diseases [29,49,50]. Our data and further validation studies could provide specific targeted clinical assays that could be developed for future clinical use. The GO and KEGG analyses in the present study showed the glycosaminoglycan/carbohydrate/inflammatory response/GTPase complex in urine, which was a consistent molecular pattern with previous hepatic tissue and serum proteomics data [32,33], indicating that the urine proteome could reflect the natural features of the human body in MAFLD. Similarly to previous metabolomic data focusing on steatosis without T2DM [11], the majority (19 of 20) of patients included in the present study were from the Hepatology Center of the Infectious Disease Department, and only one of them was clinically diagnosed with T2DM. However, GO analysis showed that insulin-like growth factor receptors have also been identified, indicating that several proteins could be identified to screen pre-diabetic patients. Moreover, it is generally well known that severe NAFLD is associated with chronic kidney disease [51]. A recent meta-analysis indicates that NAFLD is significantly associated with a ~1.45-fold increased long-term risk of incident chronic kidney disease stage ≥ 3 [52]. Urine is a direct product of the kidney, and its composition is greatly influenced by kidney function. In humans with normal kidney function, large proteins are unlikely to filter through the permselective barrier into the urine. However, the patients included in the present study were all confirmed not to have any kidney diseases, further suggesting that our results directly reflect the natural characteristics of patients with steatosis. Moreover, the results of the western blot and ELISA were normalized by urine creatinine, as described in a previous method that was applied in the investigation of kidney and prostate cancer [41,42,43]. There was a possible explanation wherein the previous study found that rats exposed to perfluorooctanoic acid showed body weight loss, significant liver swelling, reduced urea metabolism, a reduced urea concentration in urine, and an increased urea concentration in serum, in comparison with normal control rats [53]. A high urea content in serum rather than in urine may suggest that perfluorooctanoic acid exposure either decreases the ability of the liver to metabolize urea, or that urea may leak into the bloodstream due to the hepatocyte damage. Taken together, the urine molecular pattern was consistent with the molecular pattern of the hepatic tissue and serum proteomics data, indicating that urine is an important source of biomarkers.

The metabolic mechanisms leading to NAFLD reflect an imbalance of energy metabolism in the liver: excess energy, mostly in the form of carbohydrates and fat, entering the liver relative to the ability of the liver to oxidize this energy to CO_2_ or export it as very-low-density lipoproteins (VLDLs) [54]. On the other hand, glucokinase and hepatic glycogen synthesis reflect the importance of direct hepatic insulin signaling in regulating hepatic glycogen metabolism [55]; thus, the increased insulin further promotes de novo lipogenesis [54]. The class I-PI3-kinase (PI3K)–Akt/protein kinase B (PKB)–mTOR signaling pathway, through which insulin indirectly activates mTOR and suppresses autophagy, is also known to regulate glycophagy [56]. Placental factors may also promote NAFLD, including enhanced lipid and glucose transport, oxidative stress, and inflammation [57]. However, if hepatic steatosis can be reversed through metabolic interventions, then liver inflammation, liver fibrosis, and diabetes can be resolved, providing a rationale and a roadmap for the development of new strategies to address metabolic dysregulation in NAFLD and NASH [2,3,54]. In the present study, all the factors mentioned above deregulated the carbohydrate, glyco-, inflammatory response, insulin, PI3K–Akt signaling pathway, and cholesterol metabolism-related alteration, which could be detected in the urine of MAFLD patients, providing an early and non-invasive insight into the diagnosis of hepatic steatosis.

Ceruloplasmin is a copper-containing circulating protein, known as “blue substance from plasma”, which is synthesized and secreted by hepatocytes and is mostly reported in Wilson’s disease. Non-alcoholic fatty liver disease, with the lowest liver and circulating concentrations of Cu^2+^, as well as ceruloplasmin, has a more severe iron overload [58]. Genetically, ceruloplasmin variants are associated with reduced serum ceruloplasmin, which is associated with iron deposition in the liver and disease progression in patients with NAFLD [58,59,60]. Regarding the mechanism, ceruloplasmin is a multi-copper protein and it plays a critical role in iron homeostasis because it is a ferroxidase transforming noxious Fe^2+^ to Fe^3+^ and also serves as a transmembrane iron passage, allowing the release of iron to plasma Transferrin by stabilizing Ferroportin-1 [58,60,61]. However, it is still controversial that Cu and ceruloplasmin are related to clinical outcomes in patients with NAFLD and obesity. In one study, both Cu and serum ceruloplasmin were negatively associated with siderosis, ferritin, and HOMA-IR, but not liver steatosis [58]. Another cross-sectional study also found no association between the risk of NAFLD and serum Cu and ceruloplasmin levels [12]. On the contrary, some researchers found that Cu is elevated in visceral adipose tissue and liver, with little steatosis in obese patients, likely due to the concurrent increase in ceruloplasmin [62,63]. The present study showed that the urine level of ceruloplasmin, instead of serum ceruloplasmin, was associated with liver fat content, fibrosis, ALT, triglycerides, glucose, HOMA-IR, BMI, and C-reactive protein, indicating that a loss of urine ceruloplasmin is related to MAFLD. As a possible explanation, a previous study reported that the clearance of ceruloplasmin, IgG4, and IgG was significantly higher in the impaired glucose tolerance group and T2DM than in the control group [64,65,66].

Alpha-1-acid glycoprotein (AGP or ORM1) is an acute-phase protein and is majorly synthesized by hepatocytes in response to pro-inflammatory cytokines and immunomodulatory effects [67]. ORM1 was also found to be increased in T2DM [68]. ORM1 was significantly elevated in sera, liver, and adipose tissues from mice with high-fat-diet (HFD)-induced obesity and could function through leptin receptors to regulate food intake and energy homeostasis in response to nutrition status [69]. The enforced expression of ORM1 in the arcuate nucleus significantly decreased food intake, body weight, and serum insulin levels [69]. Moreover, a recent study reported that the administration of ORM1 ameliorated obesity and exerted a direct anti-fibrosis effect in adipose tissue via AMPK activation [70].

There were other highlighted proteins in the present study with future prospects for clinical translation, including 3-mercaptopyruvate sulfurtransferase (MPST), cell adhesion molecule (CADM1), angiotensin-converting enzyme (ACE), ORM1 and alpha-1-acid glycoprotein1 (ORM2), beta-ala-his dipeptidase (CNDP1), and Ras-related protein Rab-5B (RAB5B). Of 15 identifed proteins in the present study, the GO anlaysis showed that almost 40% were related to metabolic/carboxypeptidase activity, which are the keys to pathogenesis of a simple non-alcoholic fatty liver; the GO analysis showed that another 40% were related to immune dysregulation, including CADM1, Prolactin-inducible protein (PIP), and ORM1 and ORM2, of which the immune dysregulation was of liver inflammation and a progression to non-alcoholic steatohepatitis (Ref. Sheka AC et al. Jama. 2020;323(12):1175–1183). Of the 15 identifed proteins, MPST was reported to be a potential therapeutic target for NAFLD, and that the fatty acids promote fatty liver disease via the dysregulation of the MPST/hydrogen sulfide pathway (Ref. Li M et al. Gut. 2018;67(12):2169–2180). On the other hand, RAB5B, belonging to the family members of Ras-related protein, is the regulator of membrane trafficking and exosome formation, which might be related to the excretion of unidentifed proteins from donor cells (Peinado H et al. Nat Med. 2012;18(6):883–891). Nevertheless, the other 11 of 15 identifed proteins were not investigated in NAFLD, which require future studies to further investigate the underlying mechanism.

There were limitations in the present study. Firstly, the number of patients included in the present study was relatively small. Even the urine samples that are simple to collect, process, and store, the MRI-PDFF to determine the hepatic steatosis was a time-cost of 40–60 min for one participant, especially during the present pandemic time of COVID-19 causing the limited medical resource to be overwhelmed. Larger samples in a future study will likely mitigate the possible sampling bias. However, the discovered potential urine biomarkers were further confirmed by western blot and ELISA, indicating the possible medical bench-to-bed transmission. Due to the limited sample size, logistic regressions were not performed to validate whether the urine-based ORM1 and ceruloplasmin were independent biomarkers after adjusting for confounding factors, including BMI, age, transaminase, insulin resistance, etc. A future larger study would ideally address those confounding factors. Secondly, the absence of liver biopsies restricted the distinction of progressive liver diseases and represents a drawback of the present approach. However, the current study describes the first urine proteomics approach to address hepatic steatosis in MAFLD patients. Thus, the MAFLD diagnosis was more based on metabolic disorders, and steatosis was diagnosed by imaging techniques such as MRI-PDFF; a liver biopsy was not required in the design of the present study, which was based on hepatic steatosis diagnosis. More importantly, urine ORM1 and ceruloplasmin were safer to obtain, with fewer technical challenges than MRI-PDFF. Lastly, the present study was focused on hepatic steatosis, the results of fibrosis was not separately analyzed and discussed in present study. We are currently expanding our sample size and adding Magnetic Resonance Elastography (MRE) data to further investigate how urine proteomic data could diagnose fibrosis among the MAFLD patients in the future study.

In conclusion, current MAFLD and hepatic steatosis diagnoses are more based on imaging techniques or biomarkers rather than a liver biopsy. Proteomics profiling demonstrated that the molecular pattern was present in the urine samples in hepatic steatosis patients. ORM1 and ceruloplasmin are potential biomarkers to distinguish mild steatosis from healthy controls, and severe steatosis from mild steatosis, in patients with MAFLD. Urine-based ORM1 and ceruloplasmin are more accessible and non-invasive compared to blood tests; in addition, they carry no radiation risk and there is no requirement for technical expertise as with MRI-PDFF.

## Figures and Tables

**Figure 1 diagnostics-12-01412-f001:**
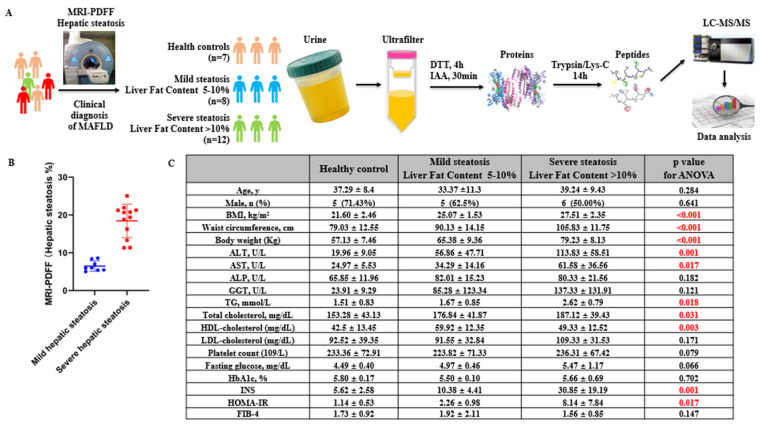
Proteomics study by using MAFLD patients’ urine samples. (**A**) The experimental design of urine profiling in proteomics for distinguishing mild/severe hepatic steatosis in MAFLD patients. Quantitative data were expressed as mean ± SD. For all laboratory measures and for continuous demographics: 1-way analysis of variance test with Bonferroni adjustments. Proportions: percentage. (**B**) Hepatic steatosis was measured by MRI-PDFF. (**C**) Baseline characteristics of included participants in discovery proteomics set according to mild hepatic steatosis (liver fat content 5–10%) and severe hepatic steatosis (liver fat content > 10%). ALT, alanine transaminase; AST, aspartic transaminase; ALP, alkaline phosphatase; GGT, γ-glutamyl transferase; TG, triglycerides; HbA1c, glycosylated hemoglobin, type A1C; INS, insulin released test; HOMA-IR, homeostasis model assessment-insulin resistance; and FIB-4, fibrosis 4 score.

**Figure 2 diagnostics-12-01412-f002:**
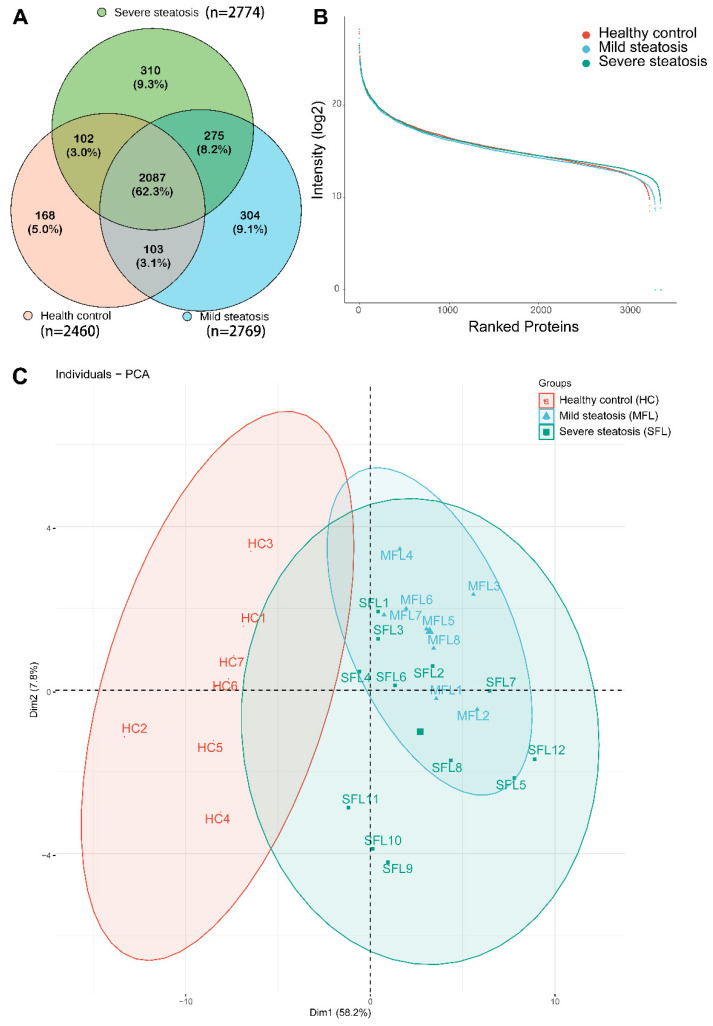
Identification and quantification of MAFLD urine samples from mild/severe steatosis and healthy controls. (**A**) The Venn diagram for the identified urine proteins from the healthy volunteers and mild and severe steatosis MAFLD patients. (**B**) The dynamic range of the intensity-based absolute quantification (iBAQ) algorithm of abundance of identified proteins from healthy volunteers and mild and severe steatosis MAFLD patients. (**C**) Principal component analysis (PCA) of urine proteome of MAFLD patients and healthy controls.

**Figure 3 diagnostics-12-01412-f003:**
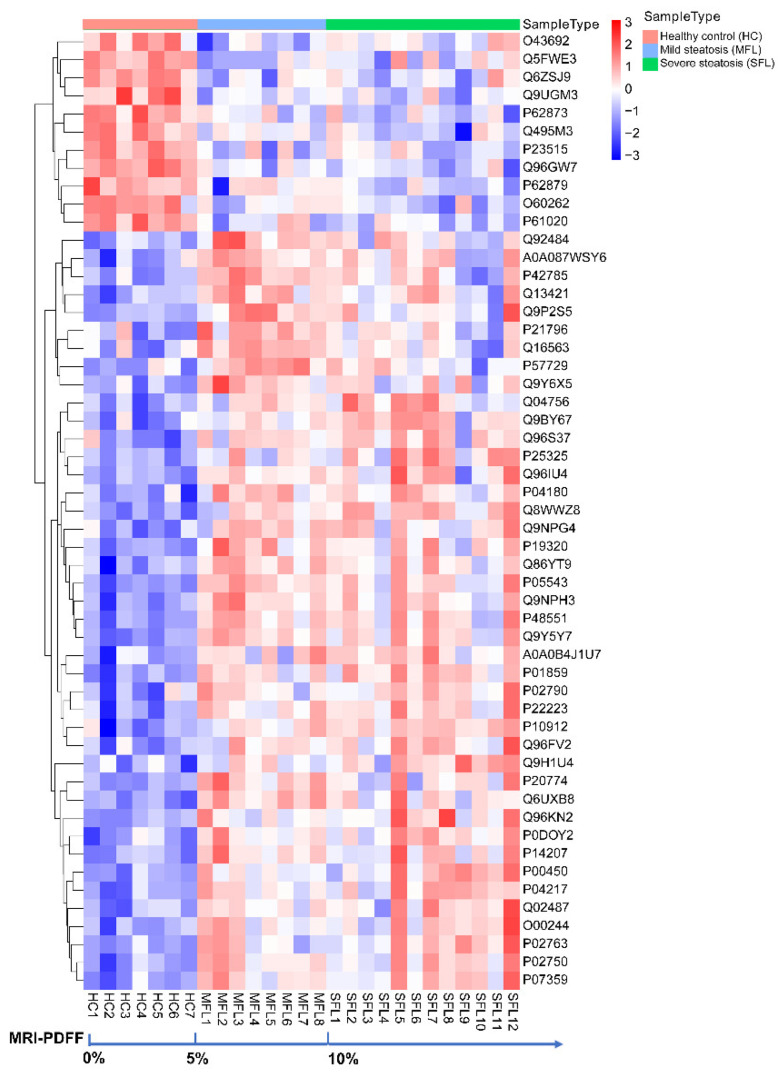
The clustering heatmap analyses differentiate mild hepatic steatosis and severe steatosis of MAFLD patients from healthy controls in proteomic features. Patients were ordered in increasing severity of hepatic steatosis measured by MRI-PDFF (from left to right). Hierarchical clustering was performed only on the 53 proteins significant by ANOVA.

**Figure 4 diagnostics-12-01412-f004:**
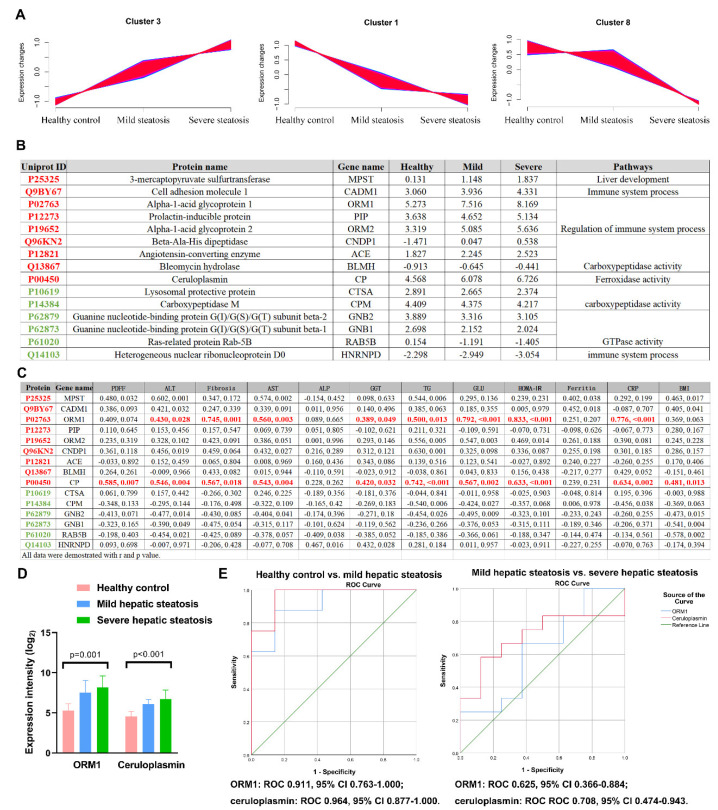
Two proteins were finally filtered for distinguishing mild/severe steatosis. (**A**) Among 12 clusters, cluster 3 included consistently up-regulated proteins from healthy controls to mild hepatic steatosis and to severe hepatic steatosis patients, whereas clusters 1 and 8 included consistently down-regulated proteins. (**B**) The GO analysis and expression (log_2_) of the filtered proteins (red color for up-regulated and green for down-regulated). (**C**) The Pearson association of 9 unique filtered proteins with clinical parameters. (**D**) The expression intensity (Un-log_2_) of selected ORM1 and ceruloplasmin, which were most strongly correlated with liver fat content (measured by MRI-PDFF), fibrosis, ALT, triglyceride, glucose, HOMA-IR, and C-reactive protein. (**E**) The diagnostic accuracy of ORM1 and ceruloplasmin for hepatic steatosis from healthy controls (right panel) and severe hepatic steatosis from mild hepatic steatosis patients (right panel) was calculated based on the expression intensity of proteomic data.

**Figure 5 diagnostics-12-01412-f005:**
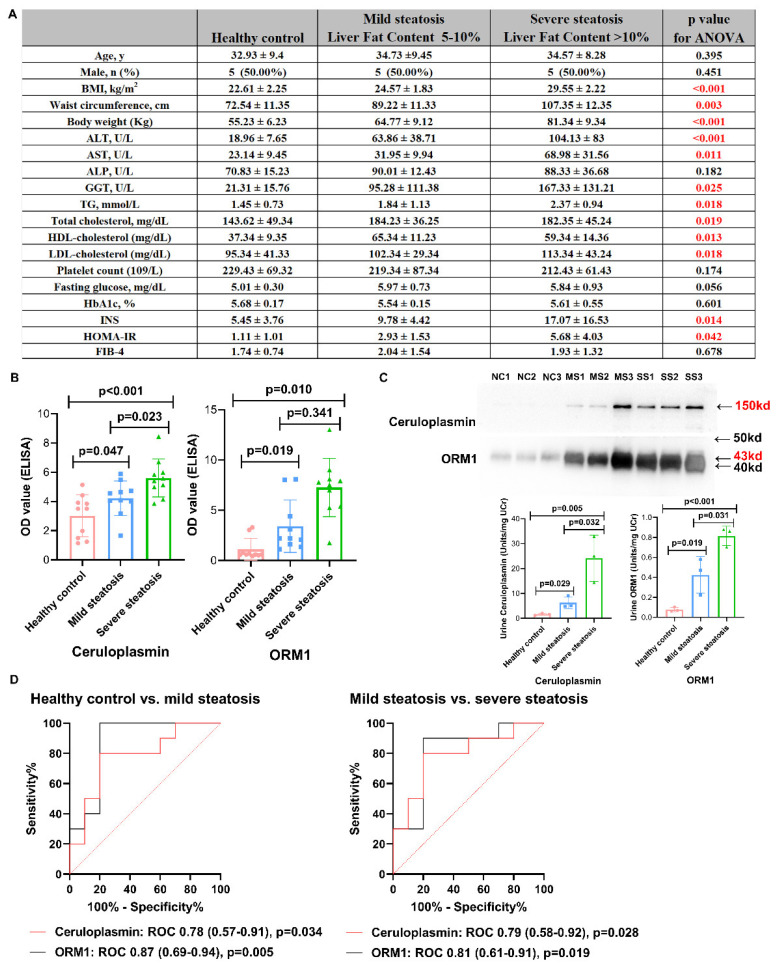
Validation of ceruloplasmin and ORMA1 in urine samples. (**A**) Baseline characteristics of participants included in the validation set according to mild hepatic steatosis (liver fat content 5–10%) and severe hepatic steatosis (liver fat content > 10%). Quantitative data were expressed as mean ± SD. For all laboratory measures and for continuous demographics: 1-way analysis of variance test with Bonferroni adjustments. Proportions: percentage. (**B**) ELISA. (**C**) Western blot; figure **below** was calculated by relative absorbance units and normalized to urine creatinine excretion. (**D**) The ROC curve (using ELISA results) for distinguishing mild steatosis and severe steatosis. NC, healthy control; MS, mild steatosis; SS, severe steatosis.

## Data Availability

The mass spectrometry proteomics data have been deposited in the Proteome Xchange Consortium via the PRIDE partner repository with the dataset identifier PXD026333. We confirm that the authors are accountable for all aspects of the work (if applied, including full data access, the integrity of the data, and the accuracy of the data analysis) in ensuring that questions related to the accuracy or integrity of any part of the work are appropriately investigated and resolved.

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
