# Peer review of "Urine Proteome in Distinguishing Hepatic Steatosis in Patients with Metabolic-Associated Fatty Liver Disease"

_diagnostics, 2022, doi:10.3390/diagnostics12061412_

Round 1

Reviewer 1 Report

Overall this is a well conceived and executed study. I have some concerns about the data analysis workflow and the discussion is lacking depth. Major points below are indicated with [Major]. A concern throughout is the highly inconsistent use of colours between figures to indicate Health/Mild/Severe steatosis.

Page 1 Line 24 (Abstract)

GO and KEGG are not defined

P1L38 (Highlights)

English here is not good.

P2L77

“has been used in several therapeutic trials as an inclusion criterion”

Only one reference given for ‘several’ trials?

P2L83

“In proteomics studies, urine represents the most frequently used biomarker in monitoring and diagnosing human diseases, due to its accessibility”

Please justify this statement with references

P2L84

“Urine biomarkers have the advantage of being less complex, with lower dynamic range and fewer technical challenges, in comparison to blood biomarkers”

This is a very definitive statement which lacks references or circumscription. I agree that the dynamic range of the Urine proteome is smaller (about 5 orders of magnitude vs 10 for serum), but both blood and urine cover enough orders of magnitude to make proteomic analysis challenging by standard proteomic approaches, and the high variability in urine concentration is a significant technical obstacle for reproducible studies. The authors are downplaying their own expertise here!

P3L104

Please clarify the number of patients and number of samples studies prominently in the first paragraph of the methods. This information should not be buried in the proteomics sample preparation section.

P3L135

“NAFLD was more defined as biopsy-proven diagnosis disease;…”

This whole sentence is not methods, and repeats the introduction.

P3L143

CAP is not defined (it is defined later, on line 164)

[Major] P4L165

There is no presentation or discussion of the fibrosis results from the FibroScan analysis – this seems to be missing (and is quite relevant!)?

P4L175

“…procedure described in the following.” is very awkward. I suggest “…procedure, as follows.”

P4L177

“10 kDa ultrafiltration tube”

Please provide supplier details

P4L182

Why did the authors digest with trypsin and then with Lys-C (implied by ‘respectively’). This is redundant! It has in the past occasionally been practice to digest with Lys-C before trypsinisation in order to improve trypsin efficiency, but digesting with Lys-C AFTER trypsin is pointless.

P4L187

“quantitative colorimetric peptide assay kit”

Please provide supplier details

P4L189

“Moreover, the digested peptides were normalized to the same concen- 189 tration at the peptide level.”

What was the concentration?

P5L218

“The datasets presented in this study can be found in online repositories. The names of the repository/repositories and accession number(s) can be found below: Proteome Xchange Consortium, accession number PXD026333.”

This is unnecessarily verbose and reads like a boilerplate statement that was pasted in last minute. I suggest:

The data presented in this study can be via the Proteome Xchange Consortium repository, accession number PXD026333

P5L222

Please provide further details of percentage of acetone–methanol solution used for precipitation, and percentage of SDS used to resolubilise (not dissolve) precipitate. Was the solution heated to resolubilise and denature?

P5L227

How was total protein normalised for loading/blotting?

P5L237

The reference given for the Jaffe reaction is for Serum, not Urine.

P5L240

“ceruloplasmin (ILC, USA)”

Unsure which company the authors are referring to here

P6L275

“Of the 69 included participants, 5 participants had to be excluded because of missing exposure or confounder data. Three participants were excluded because of a reported history of liver disease. Finally, a total sample of 67 subjects, 27 for the discovery proteomics analysis set and 30 for the validation set”

These numbers do not add up. 69 – 5 – 3 is 61 not 67. Then, 27 + 30 is not 67.

P7 Figure 1

The experimental schema in Figure 1a appears to show DTT and IAA being added before ultrafiltration. This is inconsistent with the reported methods (and does not make sense). Please correct. Also, if you are notating DTT/IAA they why not notate the digestion enzyme addition?

In Figure 1C, the numbers for males across the table are inconsistent with the sample numbers provides. If there are 5 healthy males, that cannot be 31.25% of the total healthy samples (total n of 7). and so on across the row.

Also in Figure 1C please define ALT, AST, ALP, GGT etc (all acronyms in this table)

P7L293

These data should be presented as a table; it is too dense in paragraph form.

[Major] P7L302 and Figure 2A

“The number of identified proteins in the control group, mild steatosis group, and 302 severe steatosis group grew quickly and gradually became saturated when the sample 303 size increased”

The use of ‘accumulated protein number’, ‘grew’ and ‘became saturated’ does not make sense. This was not a longitudinal study; there were three independent groups and samples were analysed in a random order. There is no concept of ‘progression’ here; the three groups were analysed simultaneously (the Healthy set was not, for example, used as a reference library for identification from the Mild set) and there is no reason to compare numbers of IDs by looking only at the number added from Healthy->Mild and Mild->Severe.

The correct comparison here is the analysis of the overlap in IDs in each set presented in Figure 2B. Figure 2A is misleading and redundant.

Accumulated protein IDs are sometimes presented in proteomics papers when comparing the effect of increasing sample size, but in these cases, firstly, data are shown for the cumulative sum for each individual added sample (not in blocks of samples) and secondly, the samples are all biological or technical replicates. I would remove Figure 2A altogether, or at minimum just show the bar chart with number of IDs per group (i.e. remove the accumulated protein number line). If you do leave Figure 2 in as a bar chart, please describe what the error bars indicate. Is it SE of number of proteins?

If the authors have free space by removing Figure 2A (and re-annotating Figure 2B->2A and 2C->2B) then an interesting figure to include as 2C would be the principal components analysis of all samples (currently Supplementary Figure 2), to show how closely the Healthy, Mild and Severe samples cluster with each other. This is a good unsupervised complement to Figure 3.

Also, in this figure, in each panel the colour schemes for Healthy, Mild and Severe are different in all three panels

P8L326 and Figure 3

Please clarify in text and legend that this hierarchical clustering is performed only on the 53 proteins significant by ANOVA analysis. The unsupervised PCA analysis (see point above) and the line “There were 53 proteins identified with 331 significantly different expression among the three groups (Supplementary Table S1)” should come before Figure 3 is introduced.

P10L344

“, 2087 commonly identified proteins” should read “, _the _2087 commonly identified proteins” to make it clear this is the complete shared set (not an arbitrary selection of 2087).

P10L345 and Supplementary Figure 5

How was the choice of number of clusters made? Also, what does the blue-yellow-red gradient at the bottom left of Supplementary Figure 5 indicate? The axes are unlabelled and there is no title.

Also, there are typos on the x axes; ‘Mild steaosis’ and ‘Several steatosis’

[Major] P10L350 and Figure 4

“Combining the filter results and the results of the 53 significantly changed urine proteins in the previous analysis”

How? There are no details on this combination in methods.

This part of the analysis does not make sense. The authors have already identified the significantly varying proteins via ANOVA; to discover which proteins are different between specific Healthy/Mild/Severe pairings, they should use post-hoc testing. Why are they now going back to the full set of shared proteins to perform cluster analysis and use that as a filtering step? This is redundant and statistically unjustified.

The authors could show how many statistically different proteins are in each MFuzz cluster and use that logic to decide which of the 53 significant proteins merit further investigation. This does not appear to be what has been done.

Figure 4D & E, the colours corresponding to severity (blue-red-green) are the opposite to those corresponding to severity in Figure 3 (green-red-blue).

P10L356

“Moreover, among these 16 proteins, Pearson correlation analysis found that there were 2 of 9 unique proteins”

Where did the 9 come from? Do the authors mean “2 of the 9 upregulated unique proteins”. Also, 9 + 6 does not equal 16?

P13L400 (Figure 5)

Figure 5B & C, the colours corresponding to severity are different yet again

Figure C – No loading controls on Western, some samples not shown. Why?

P13L405 Figure 5 Legend

Typo: ‘figure blow’

P14L408

The discussion is incomplete. It needs more work to tie the highlighted proteins to the wider context and the results of the GO analysis. It would also benefit from more discussion of future prospects for translation (especially of the 7 other significant proteins which are not discussed at all?)

Is there any data to suggest whether the observed results are correlated to liver steatosis specifically, or just BMI more generally? This is not discussed, and seems like an omission (a larger study would ideally regress out BMI in the analysis by use of a 2-way ANOVA).

P14L452

“indicating that urine is a biomarker that may be easier to obtain than serum”

Urine is not a biomarker! It is a source of biomarkers.

Author Response

Thank you very much for reviewing our manucript. Please find the response letter in the attached files.

Reviewer 2 Report

The authors conclude that Ceruloplasmin and ORM1 are potential biomarkers in distinguishing mild and severe steatosis in MAFLD patients. Even if discovery set and validation set do not have adequate numerosity conclusions on urine samples that are simple to collect, process, and store, seems  to be strong 

No data on  Liver fibrosis are reported discuss on this point

Author Response

Thank you very much for reviewing our manuscript. Please find the response letter in the attached files.

Round 2

Reviewer 2 Report

The authors modified text, addressing quite all questions